# Visual analysis of geographical distribution of poets in Song China based on *Complete Song Poetry*

**Enhai Lei**[1,2]* , **Xudong Hu**[1,2]

1 School of Chinese Languages and Literatures, Lanzhou University, Lanzhou, P. R. China, 2 Center for Chinese Studies, Lanzhou University, Lanzhou, P. R. China

☯ These authors contributed equally to this work.

* leieh@lzu.edu.cn

**Data Availability Statement:** All relevant data are within the manuscript and its Supporting Information files.

**Funding:** This research was funded by the Gansu Education Department (Grant No. 2022CYZC-09).

## Abstract

This article analyzes the geographical distribution of poets in Song China based on *Complete Song Poetry* (Quansongshi 全宋詩), which includes poems from over 9000 poets, with 6056 of them having clear native place. Visualization strategy is used to present the geographical distribution of these 6056 poets, and its formation factors are also analyzed. The results reveal that majority of the poets come from Zhejiang, Henan and Sichuan provinces, with Guangdong province also exceeding a hundred poets despite its remote location. In the central area of Ningbo, Luoyang and Meishan, there are also large distribution of the poets. Over time, the density of poets in the south increases while it decreases in the north, and this pattern becomes more pronounced as time progresses. By employing visualization strategy, we aim to provide a more comprehensive understanding of the geographical distribution of poets of *Complete Song Poetry* in Song Dynasty to reflect the cultural, economic, and political development of Song China.

## Introduction

China is a country of poetry, and after the thriving of poetry in Tang Dynasty (618–907), the Song China (960–1279) saw another summit in the development of Chinese poetry. Thousands of poets appeared during this time and scattered across China. The geographical distribution of poets is closely related to the regional social development. Consequently, studying on the geographical distribution of poets in Song Dynasty can get insights into the cultural, economic, and political development of Song Dynasty society. From statistical and geographical distribution perspectives, the research works on the distribution of poets in the Song Dynasty have been performed previously in the academics, resulting in many research achievements. Up to now, there are two research methods used to study the geographical distribution of poets in Song Dynasty. One is to investigate the geographical distribution of poets from a statistical perspective, while another tries to present it in visual methods. For instance, as far back as in the 1990s, Zeng Daxing's book "Geographical Distribution of Chinese Literary Masters of Previous Dynasties" [1] presented a statistical analysis of the geographical distribution

Enhai Lei received the funded project. The funders had no role in study design, data collection and analysis, decision to publish, or preparation of the manuscript. Website of the funder: http://jyt.gansu.gov.cn/.

**Competing interests:** The authors have declared that no competing interests exist.

of ancient Chinese literary masters. In chapter 6 of this book, the geographical distribution patterns of literary scholars in Song, Liao, and Jin Dynasties along with their formation reasons were discussed based on the number of writers' native places recorded in Tan Zhengbi's "Dictionary of Chinese Literati" [2], and 1168 poets were involved in total. Zeng's work simply analyzed the geographical distribution of Song poets statistically in the form of tables, and examined the correlation between the distribution of poets and the local culture. Afterwards, Wang Zhaopeng and Liu Xue published a paper in 2003, entitled "Statistical Analysis of Poets in Song Ci" [3], in which a statistical analysis of 880 poets in Song Dynasty was performed to present their geographical and temporal distributions. They concluded that the development of Song Ci, a special type of poem in Song Dynasty, was intimately linked to the prosperity of Song poetry. Comparing to Zeng's research work, Wang and Liu's work, combining the economy, politics, and population factors of the Song Dynasty, provides a more detailed geographical distribution of poets in the Song Dynasty. Additionally, Wang Xiang's "The Geographical Distribution of Poets in the Northern Song Dynasty and Its Literary Significance" [4] in 2006 recorded the geographical distribution of 975 poets in the Northern Song Dynasty based on *Complete Song Poetry* [5], which was compiled by the Center for Ancient Chinese Classics and Archives of Peking University and published by Peking University Press in 1998. He divided the Northern Song Dynasty into four stages from a literary history perspective to discuss the geographical distribution of poets, providing insights into the literary prosperity and cultural center of the Song Dynasty. In 2020, Lou Xinxing visually presented the distribution of poets in *Complete Song Ci* at the provincial and county levels in her article "Geographical Distribution of Authors in *Complete Song Iambic Verse Collection* and the Cause" [6], with particularly focusing on poets from Zhejiang province. She is one of the few scholars in recent years to utilize visualization methods to study the geographical distribution of poets in Song Dynasty. She has drawn several maps, and intuitively and clearly show the geographical distribution of poets in the Song Dynasty. However, only 820 Song poets were involved in her research work, which did not cover the majority of poets in Song Dynasty.

Overall, scholars have achieved significant research results on the geographical distribution of poets in the Song Dynasty in terms of statistics. However, studies employing visualized presentations of the poet's distribution are relatively scarce, and the poets involved in such research works are only around a thousand people, which did not fully cover the poets in Song Dynasty.

## Research objectives, data collection and processing, and research method

### Research objectives

Building on the above findings, the previous studies on the geographical distribution of poets during the Song Dynasty involved about a thousand individuals. In fact, the *Complete Song Poetry* documents over 9000 poets from the Song Dynasty in its total of 72 volumes, almost encompassing all of the poets in Song Dynasty and offering a comprehensive overview of Song Dynasty poets. Therefore, studying on the geographical distribution of all poets based on the *Complete Song Poetry* is essential for a comprehensive understanding of poets' distribution in Song Dynasty, and then offering more accurate insight into the regional characteristics of literature in Song Dynasty. In this work, we will perform a visualized analysis of geographical distribution of all Song poets with clear native places based on the *Complete Song Poetry*, and discuss its formation reasons to provide a comprehensive understanding of geographical distribution of Song poets and its relation with the development of Song Dynasty society.

## Data sources and processing

The *Complete Song Poetry* contains over 250,000 Song poems. Due to the massive number of poems, the Department of Chinese Language and Literature of Peking University developed the *Complete Song Poetry Analysis System* [7] in 2005, which support full text search, poet biography search and advanced search. This system will be instrumental in collecting and processing the extensive and intricate data of poets with clear native place records, enabling quantitative analysis of their geographical distribution through statistical method. This system ensures the accuracy of the data we obtain, and a total of valid 9200 poet's data were collected from the system, with 6,056 poets having clear known native place and 3,144 poets having no clear native place, which provides a basic data guarantee for our subsequent research work.

To achieve data visualization, we use Tianditu (Map World) [8] to obtain accurate latitude and longitude coordinates for each location of poet's native place. Furthermore, the data of 6,056 poets with clear known native place was imported into GIS software to present the geographical distribution of poets.

## Research method

**Spatial query and measurement.** Spatial query, one of the basic functions of GIS, allows users to filter and query spatial objects based on geographic location, attributes, and other conditions. Through spatial queries, data about the corresponding objects can be obtained for further detailed analysis. Spatial measurement is to measure the spatial attributes of geographic objects, which allow us to better understand the spatial distribution characteristics and relationships of geographic elements. This study will utilize spatial query and measurement to spatialize the collected data of poets' native places to present their distribution on the map.

**Jenks natural breaks optimization.** This method aims to determine the best arrangement of data into different classes, which is particularly effective for classifying data with uneven distribution. In this study, Jenks natural breaks optimization has been used to achieve effective stratification for the massive data of poets' native places.

**Graduated symbols renderer.** The graduated symbol renderer is a sophisticated tool for representing quantitative data, excelling in categorizing quantitative data into ordered classes, with each class distinguished by a consistent set of symbols. The system applies a series of graduated symbols to each class, incrementally enhancing the visual representation. This study applies the graduated symbol renderer to process visually the data of poets' native place, enabling it to be presented more accurately on the map and allowing for a more intuitive understanding of the poets' geographic distribution.

## Results and discussion

### The recordings of poets' native place in *Complete Song Poetry* and their characteristics of geographical distribution

The poets' native place recorded in *Complete Song Poetry* follows the following formats: first, the place names at that time are recorded, meanwhile the current place name or the current provincial administrative region is indicated. For examples, "Liang Zhouhan (929–1009), with the courtesy name Fubao, was born in Guancheng, now Zhengzhou, Henan; or Li Fang (925–996), with the courtesy name Mingyuan, was born in Raoyang, Shenzhou, now part of Hebei Province." Secondly, when the native place is recorded, the place where the poet lived or moved to is also indicated. For instance, "Yang Wei (1106–1156), with the courtesy name Fuguang, was born in Jingling, now Changzhou, Jiangsu. After moving southward, he lived in Shengxian, now Zhejiang." Thirdly, if poet's native place has multiple explanations, each case

is recorded one by one. For example, "Du Qi (1005–1050), with the courtesy name Weichang, was said to be from Jinling, now Nanjing, Jiangsu Province. It is also said that he came from Wuxi, now in Jiangsu Province." In addition to Chinese poets, some poets from Korea and Japan were also recorded in *Complete Song Poetry*. Among them, three poets were from Korea, namely Park In-yang (Jukju or Pyeongju, Korea), Park Kyung-so (Jukju, Korea) and Lee Ja-ryang (Imju, Korea). All three of them were Korean envoys. One poet is from Japan, a Japanese monk named Shakkaku Ajōnin. When the poet's native place is not clear, the *Complete Song Poetry* will record them in following manners: first, the phrase "no biography" will be used to indicate that the poet's situation is unknown, and there are about 1,121 poets with "no biography" recorded in *Complete Song Poetry*. Another way to record the poets is to directly describe the poets' deeds. For example, "In the second year of Emperor Taizu Gande reign (964 AD), Yingzhi participated in and passed the examination of Xianliang Fanfzheng Zhiyan Jikjian Ke (an examination on virtuous and upright officials giving candid advice) and was awarded Mishusheng Zhuzuolang (an assistant officer in the secretariat). He also served as the magistrate of Luoyang County." The final way is to record the era in which the poet lived, or to record the people who lived with the poet in the same era. For example: "Qian Sheng, lived in Emperor Renzong's era, or Yang Zhixiang, contemporary with Li Zhou".

Based on the above regulations, the analysis conducted by the *Complete Song Poetry Analysis System* reveals that there are 9,200 poets associated with the *Complete Song Poetry*. Among them, 6,056 poets, including 3 poets from Korea and 1 poet from Japan, have clear known native place, while 3,144 poets have no clear native place. To investigate the specific characteristics of the geographical distribution of these poets within the *Complete Song Poetry*, excluding 4 poets from Korea and Japan, a provincial statistics detailing the native places of the poets has been shown in Fig 1A and 1B, which utilizes the current provincial administrative divisions and is supplemented by the Specific Statistics of the Native Places of the Top 50 Poets in the *Complete Song Poetry*, in which only the top 50 regions are listed in Fig 2 due to too much data involved.

According to Fig 1, the provincial distribution of the poets' native places in the *Complete Song Poetry* can be divided into three echelons based on population number. The top tier provinces consist of provinces where the population proportion exceeds 8%, including Zhejiang, Fujian, Jiangxi, and Jiangsu provinces. These four provinces are all located in the southeast coastal areas, with a total of 4060 poets. Zhejiang province leads the tier with 1,562 individuals, making up 25.81% of the provincial administrative region statistics, significantly higher than the other provinces. This means that one in every four poets with known native places in the *Complete Song Poetry* comes from Zhejiang province. Fujian province ranks the second with 1098 people. These two provinces are the only ones with over a thousand poets. Jiangxi and Jiangsu provinces closely follow behind with 823 and 577 individuals, respectively. The total proportion of the poets in these four provinces, account for 67.08% of all Song poets. It can be concluded that over half of the poets with a clear native place in the *Complete Song Poetry* stem from these four provinces. As further suggested by the data in Fig 1, among the top ten provinces, the eight provinces, including Zhejiang, Fujian, Jiangxi, Jiangsu, Sichuan, Anhui, Hunan and Guangdong, are all from the southern regions, indicating that the main creative force of Song poetry are from southerners.

The second tier includes the provinces with the population contribution ranging from 2% to 8%, including Henan, Shandong, Sichuan, Anhui, Hunan and so on. The total number of poets in this echelon is 1,496, which is far behind the first echelon. Notably, the poets in this tier are mainly concentrated in the Henan and Shandong provinces.

Further examination of the data in Fig 1 reveals an interesting geographic distribution phenomenon: Henan and Shandong are the sole northern provinces among the top ten provinces.

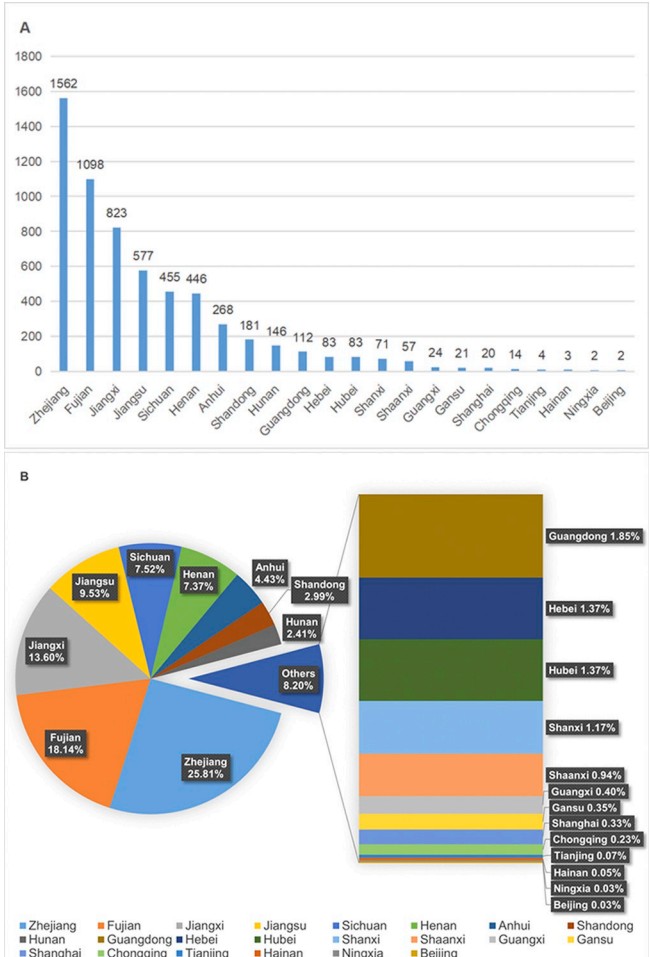

**Fig 1. Provincial statistics of poets' native places in *Complete Song Poetry*.** (A) Bar chart based on provinces. (B) Pie chart based on percentage. This figure shows the ranking of the poets' geographical distribution on provincial level as well as the percentage of poets in each province.

Furthermore, the poet population in Henan are approximately 2.4 times that of Shandong, indicating that the northern epicenter of Song poems was overwhelmingly based in Henan, with Shandong playing a complementary role.

Finally, the third tier comprises provinces with a poet population less than 2%. This group includes Guangdong, Hebei, Hubei, Shanxi, Shaanxi, Guangxi, Gansu, Shanghai, Chongqing, Tianjin, Hainan, Ningxia, Beijing, and so on. These provinces, while contributing less numerically to the poetic corpus of that time, still played no negligible role in the rich tapestry of Song poetic culture.

The data presented in Fig 2 offers valuable insights into the geographical distribution of poets at the county and city levels. It is evident that the most outstanding place is Kaifeng, Henan, which ranks first in terms of the number of poets at county and city levels, despite the southeastern coastal areas being the main source of poets. This emphasizes the significance of Kaifeng in the literature of the Song Dynasty. Moreover, Fig 2 highlights that Kaifeng and Luoyang in Henan Province are the only northern cities among the top 50 regions, further solidifying the notion that the creative epicenter of the northern region is concentrated in Kaifeng and Luoyang. On the other hand, Chengdu and Meishan in the southwest region rank 12th

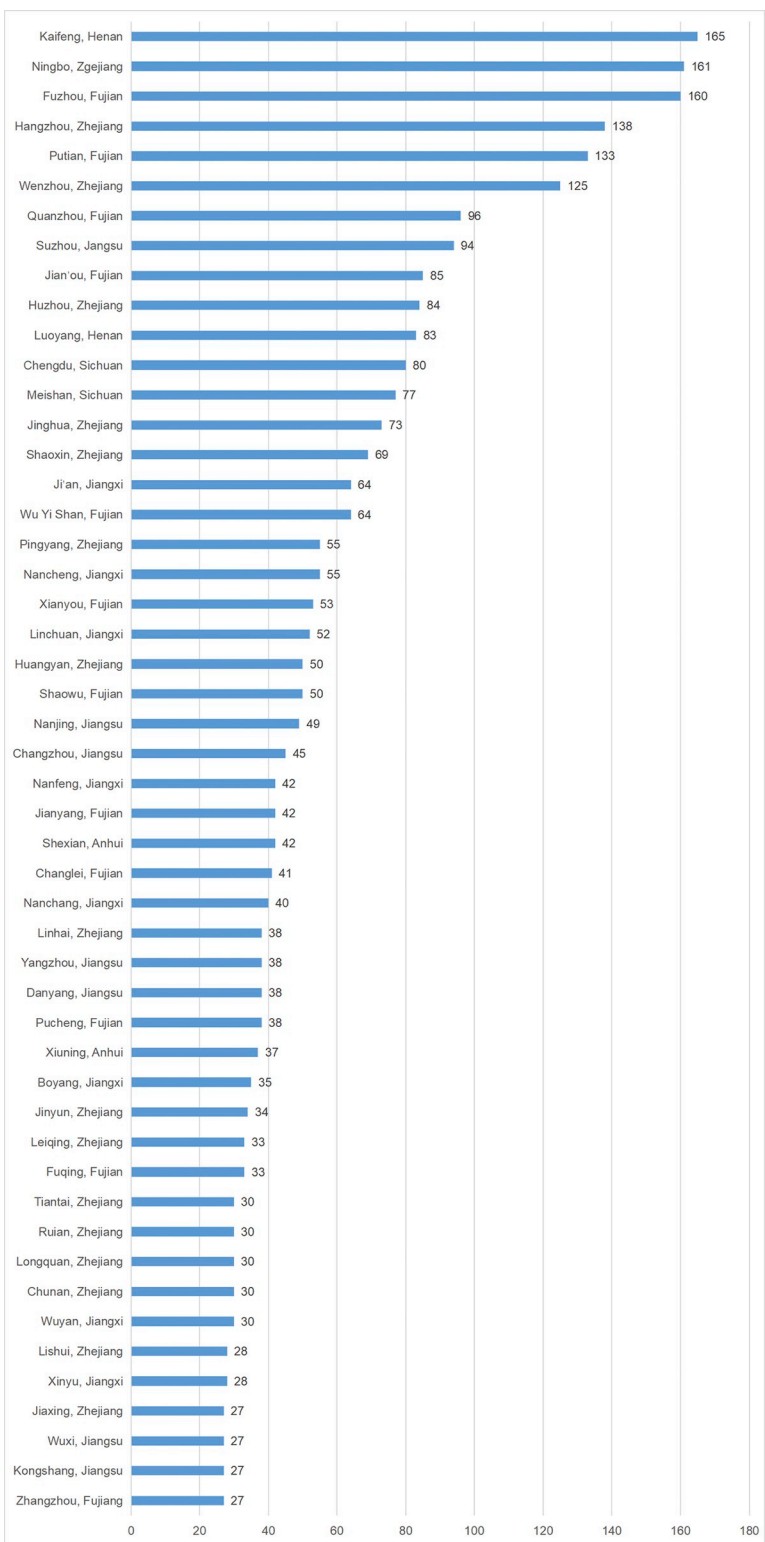

**Fig 2. Statistics of native places of top 50 poets in *Complete Song Poetry* at the county and city levels.** Due to the massive data, this figure only lists the top 50 specific locations. The entire data see S1 Dataset.

and 13th at the city level with 80 and 77 individuals respectively, indicating that they are the creation hubs of the southwest region.

To more vividly present the distribution patterns of the poets in the *Complete Song Poetry*, the data gathered above was put into GIS software, allowing for the creation of density distribution maps of the poets within the *Complete Song Poetry* (Fig 3A), as well as segmented density distribution maps of the poets in the northern areas (Fig 3B), the southwest areas (Fig 3C) and the southeast coastal areas (Fig 3D) In these distribution maps, the circles serve as distribution markers, with the size and color depth of the circles indicating the distribution density of the poets. The larger the circles and the darker the color, the more the poet distributions are. From those distribution maps, it becomes evident that the poet distribution in both quantity and space in the *Complete Song Poetry* exhibits the following characteristics.

Firstly, as presented in Fig 3A, the geographical distribution of the poet's native place in the *Complete Song Poetry* generally presents the "three cores" pattern. The first and biggest one is the "Southeast Coastal Core" (Fig 3D), predominantly comprising Zhejiang, Fujian, Jiangxi, and Jiangsu. The majority of poets in this core are born along the coastal line, which also exhibits the highest population distribution with a well-balanced population in each region. The second is the "North Central China Core" (Fig 3B), which includes mainly Kaifeng and Luoyang, with the number of poets gradually decreasing from Kaifeng and Luoyang away to other regions, establishing dual centers in the north. The last, the "Southwest Inland Core" (Fig 3C), features Chengdu and Meishan as the dual centers, predominantly concentrated within the Sichuan Basin independently. These three cores encompass most of the poets' native places and present the main sources of the poets in the Song Dynasty.

Secondly, the core of the southeastern coastal area roughly presents a strip-like distribution feature along the vertical and horizontal axes, with the intersection point of the vertical and horizontal axes located in Hangzhou. The vertical axis starts from Yangzhou and Nanjing, heading south through Zhenjiang, Changzhou, Suzhou, Huzhou, Hangzhou, Shaoxing, Ningbo, Taizhou, and Wenzhou, and ends at Fuzhou after passing Putian, Quanzhou, and Zhangzhou in Fujian. The horizontal axis starts from Hangzhou and divides into two lines from east to west, i.e. north line and south line. Both lines end in Ji'an of Jiangxi province. The northern line passes through Shexian and Xiuning in Anhui province, and then through Wuyuan, Poyang, Nanchang, Xinyu in Jiangxi province. The southern line passes through Dongyang, Yiwu, Jinhua, Quzhou in Zhejiang province, and then through Pucheng, Jian'ou, Wuyi Shan, Shaowu in Fujian province, and Nancheng, Nanfeng, Fuzhou, Jishui in Jiangxi province. The southern line covers more areas than the northern line, with more poets born in this region.

In summary, the geographical distribution of poets in the *Complete Song Poetry* varies greatly across different regions. The distribution in the south is much denser than that in the north, meanwhile the distribution in the east is denser than that in the west. In the southeastern coastal regions, the distribution is concentrated around clusters of cities and displays a strip-shaped pattern. In the north and southwestern regions, the distribution centers in each region are composed of twin cities and present a scattered distribution.

## The formation reasons behind the geographical distribution characteristics of poets in the *Complete Song Poetry*

Generally speaking, many factors such as local favorable natural conditions, convenient transportation, high population density, a developed economy and a strong cultural atmosphere have a positive influence on the distribution of the number of poets.

Taking Kaifeng and Hangzhou as examples, they are ranked first and fourth in Fig 2, respectively. Since it was established as the Northern Song capital in 960 AD, Kaifeng became

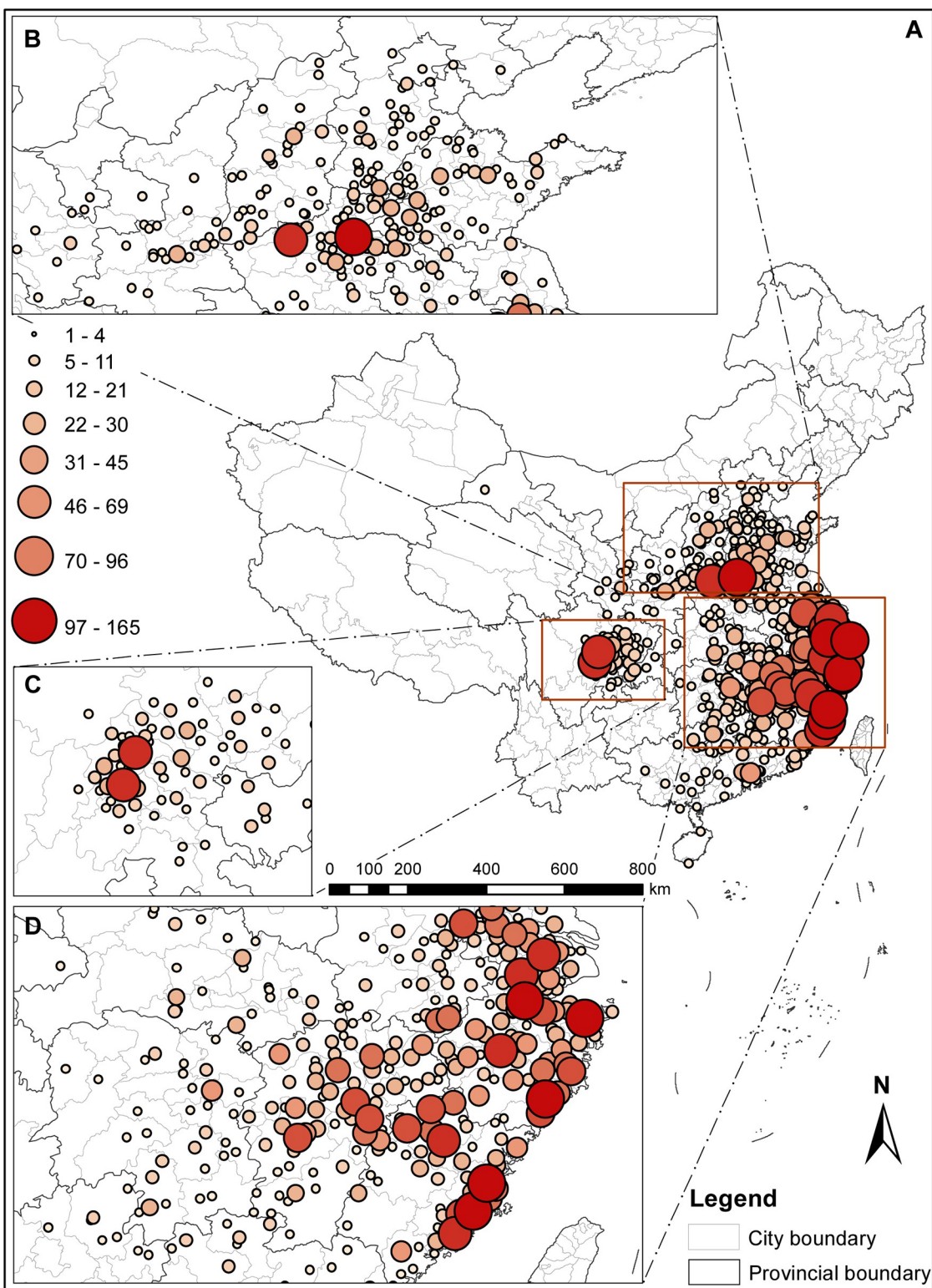

**Fig 3. Geographical distribution map of density of poets' native places in the *Complete Song Poetry*.** (A) Distribution in entire country. (B) North Central China Core. (C) Southwest Inland Core. (D) Southeast Coastal Core. Coordinate information of the poet's native place see S1 Dataset. The base map was obtained from Natural Earth [9]. Map credit: Xudong Hu.

a combination of imperial space and commercial space, gathering a large amount of wealth [10]. Many high-ranking imperial officials who were also outstanding poets lived here. After Kaifeng was captured by the Jin army in 1127 AD, a large number of royal members and scholar-officials fled south and established a new capital in Hangzhou. Patriotic poets also gathered in this city with the magnificent scenery of West Lake, and accused the fall of the north and the inaction of the authorities with poetry [11]. On the contrary, taking Hainan as an example, it was a remote place used to exile prisoners during the Song Dynasty. Therefore, few poets were distributed here.

In previous studies, scholars have also explored the reasons behind the distribution patterns of poets in the Song Dynasty. For example, Wang Zhaopeng and Liu Xue analyzed the statistical data of Song Ci poets in their "Statistical Analysis of Song Ci Poets", [3] and found that the distribution of poets correlates with the densely populated areas of the Song Dynasty. The article "Geographical Distribution of Authors in *Complete Song Iambic Verse Collection* and the Cause " [6] by Lou Xinxing investigated the geographical distribution of poets in Zhejiang, Jiangxi, and Fujian provinces, and highlighted the impact of various factors such as politics, economy, culture and education on the distribution of poets in these regions. Herein, we gather the data of the poets in the "*Complete Song Poetry*" in more detail and conduct visual analysis, which not only support the above conclusion, but also reveal some new characteristics at both the macro and micro levels that have not been examined in previous studies. For example, despite Guangdong province being in the third tier with the proportion of poets being less than 2%, the number of poets in this region still exceeds 100 people, surpassing other regions in the same tier (Fig 1). This is a clear difference between Guangdong and other regions. At the county and city levels, Ningbo, although not being the provincial capital city, ranks the second in the number of poets, trailing only behind Kaifeng and even surpassing its capital city Hangzhou. Luoyang in Henan province, alongside Kaifeng, serves as dual distribution cores in the northern region, despite being ranked 11th in the number of poets, lower than that of southern regions. Meanwhile, Meishan in Sichuan, although situated in a remote southwestern region, boasts a number of poets comparable to that of southeastern counties and cities. Therefore, the factors influencing poet distribution in Guangdong at the provincial level, and that in Ningbo, Luoyang and Meishan at the county and city levels, will be further explored here with the help of visualizing analysis to further enhance our understanding of the distribution of poets in the Song Dynasty based on *Complete Song Poetry*.

**The distribution of Guangdong poets and its formation causes.** To present the detailed distribution of poets in Guangdong province, this article further organizes the distribution data of Guangdong poets. Due to the differences in geographical evolution between ancient and modern times, the native places of poets in the *Complete Song Poetry* are not completely consistent with the administrative divisions of the Song Dynasty. To facilitate the statistics, the following provisions are made: Firstly, the poets, whose native places are in Guangdong but whose specific states or counties are unknown, will not be counted. Secondly, the statistics of the poets' native places are based on the county. If the specific county of their native place is unknown, the statistics will be based on the state capital. Following on these provisions, a distribution of Guangdong poets in *Complete Song Poetry* was created, as shown in Table 1 and Fig 4. The data are imported into GIS software again to generate a visual geographical distribution map of Guangdong poets in *Complete Song Poetry*, as illustrated in Fig 5.

According to Table 1, Figs 4 and 5, it is evident that Guangzhou is the central hub of poets in Guangdong province, with a significant 44 poets, far exceeding other states and counties, securing the first position. The primary reason behind this phenomenon is the strong economic foundation laid by Guangzhou's flourishing maritime trade during the Song Dynasty. Due to the complete loss of the old territories of the Han and Tang Dynasties, the northern

**Table 1. Distribution of Guangdong poets in the *Complete Song Poetry*.**

| State | County | Number of poets | Total Number |
|---|---|---|---|
| Guangzhou | Dongguan | 25 | 44 |
| | Guangzhou | 12 | |
| | Zengcheng | 3 | |
| | Shunde | 2 | |
| | Zhongshan | 1 | |
| | Nanhai | 1 | |
| Shaozhou | Qujiang | 11 | 14 |
| | Lechang | 2 | |
| | Shaoguan | 1 | |
| Chaozhou | Chaoyang | 5 | 13 |
| | Chaozhou | 5 | |
| | Huilai | 1 | |
| | Jieyang | 1 | |
| | Chao'an | 1 | |
| Lianzhou | Lianzhou | 9 | 11 |
| | Lianshan | 2 | |
| Huizhou | Boluo | 3 | 7 |
| | Huizhou | 2 | |
| | Guishan | 1 | |
| | Heyuan | 1 | |
| Yingde | Yingde | 6 | 6 |
| Nanxiong | Nanxiong | 5 | 5 |
| Meizhou | Meixian | 3 | 3 |
| Deqing | Deqing | 2 | 2 |
| Yangjiang | Yangjiang | 1 | 1 |
| Xinzhou | Xinxing | 1 | 1 |
| Suixi | Suixi | 1 | 1 |
| Shenzhen | Shenzhen | 1 | 1 |
| Xunzhou | Longchuan | 1 | 1 |
| Xingning | Xingning | 1 | 1 |

If the names of the state and county are the same, it indicates that the distribution of some poets can only be accurate to the state.

territories of Song Dynasty frequently faced the threat of wars. In order to ensure a stable source of revenue through taxation, the Song government paid great importance to foreign maritime trade, and made many liberal policies to promote foreign trade [12].

Guangzhou, with its convenient sea transportation, emerged into a vital trading port. The government actively fostered continuous trade exchanges with many countries in Southeast Asia and the Indian Ocean, such as Samboja, India, and Arab countries. In 971 AD, Guangzhou Shibosi, i.e. the Guangzhou Maritime Trade Supervisorate, was established. It was the earliest municipal shipping department established in the Song Dynasty, highlighting the significant importance of Guangzhou's trade status at that time. Prosperous overseas trade laid a strong economic foundation for Guangzhou, making its cultural customs comparable to those of the Central China. These are also clearly recorded in "Difangzhi", which is also translated as "local gazetteers" [13]. In the following, we will present some formation reasons about geographical distribution of Guangdong poets in Song China illustrated by "local gazetteers" and other related historical records.

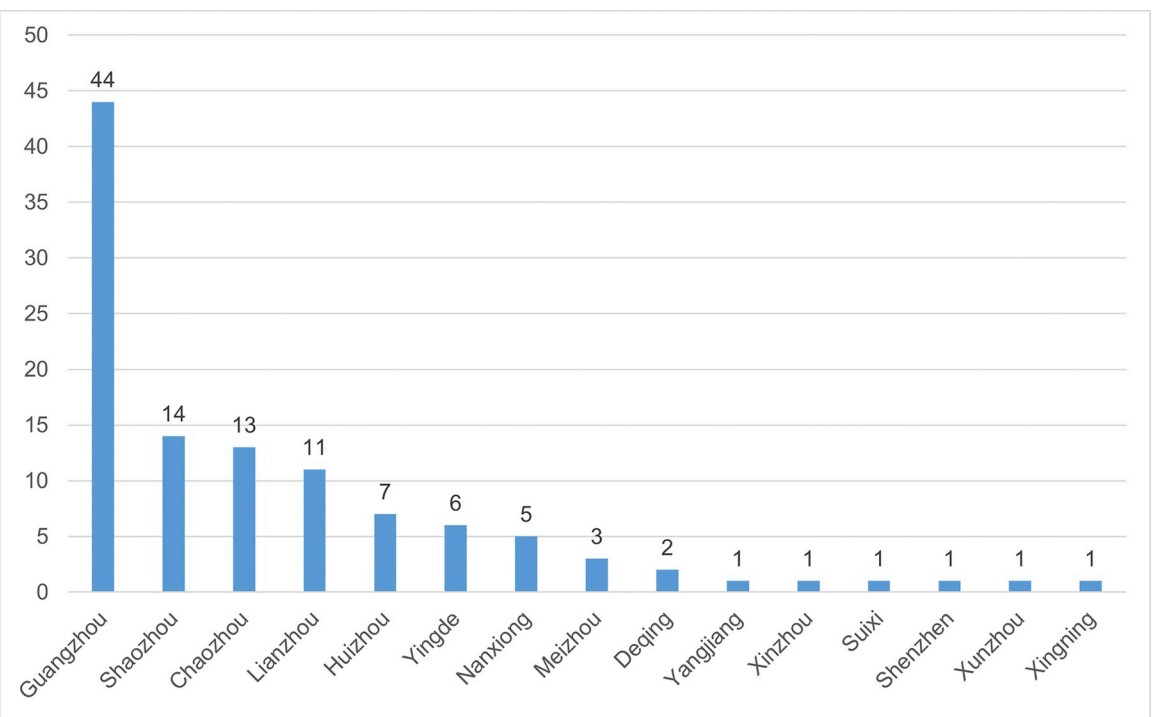

**Fig 4. State distribution of poets of Guangdong in *Complete Song Poetry*.** The detail data see S2 Dataset.

According to the records by scholar-official Li Maoying (1201–1257) in Southern Song Dynasty:

"Guangzhou has always been known by its abundance in natural resources. . . massive bananas and nuts are grown here, a large amount of shrimps and crabs are raised here. Meanwhile, Schools and temples are scattered here. Locals are willing to have good teachers to educate their children. The sound of strings and reciting poems complement each other. Although the number of talented people participating in the imperial examination is less than that in the Central China, the number of people passing it is the same as in the Central China" [14].

Among the various regions in Guangzhou, Dongguan has the largest distribution number of poets. Wang Zhongxing (1158–1210), a scholar-official in Southern Song Dynasty, stated that:

"Dongguan was initially a county and later transformed into a center for supervising county governance, a model role that persists to this day. The land here is fertile, and the number of talented people with sophisticated farming skills, war skills, and famous official positions surpassed other places" [15].

In addition to Guangzhou, Shaozhou, Chaozhou as well as Lianzhou also ranked second, third, and fourth respectively. The total number of poets in each region is all above 10. Shaozhou, adjacent to the Xiang River, has been an important transportation hub since ancient times. Qujiang in Shaozhou is full of talented people. Li Bo (773–831), a Tang Dynasty official, praised this as: "Qujiang in Shaozhou is the hometown of generals and ministers" [16].

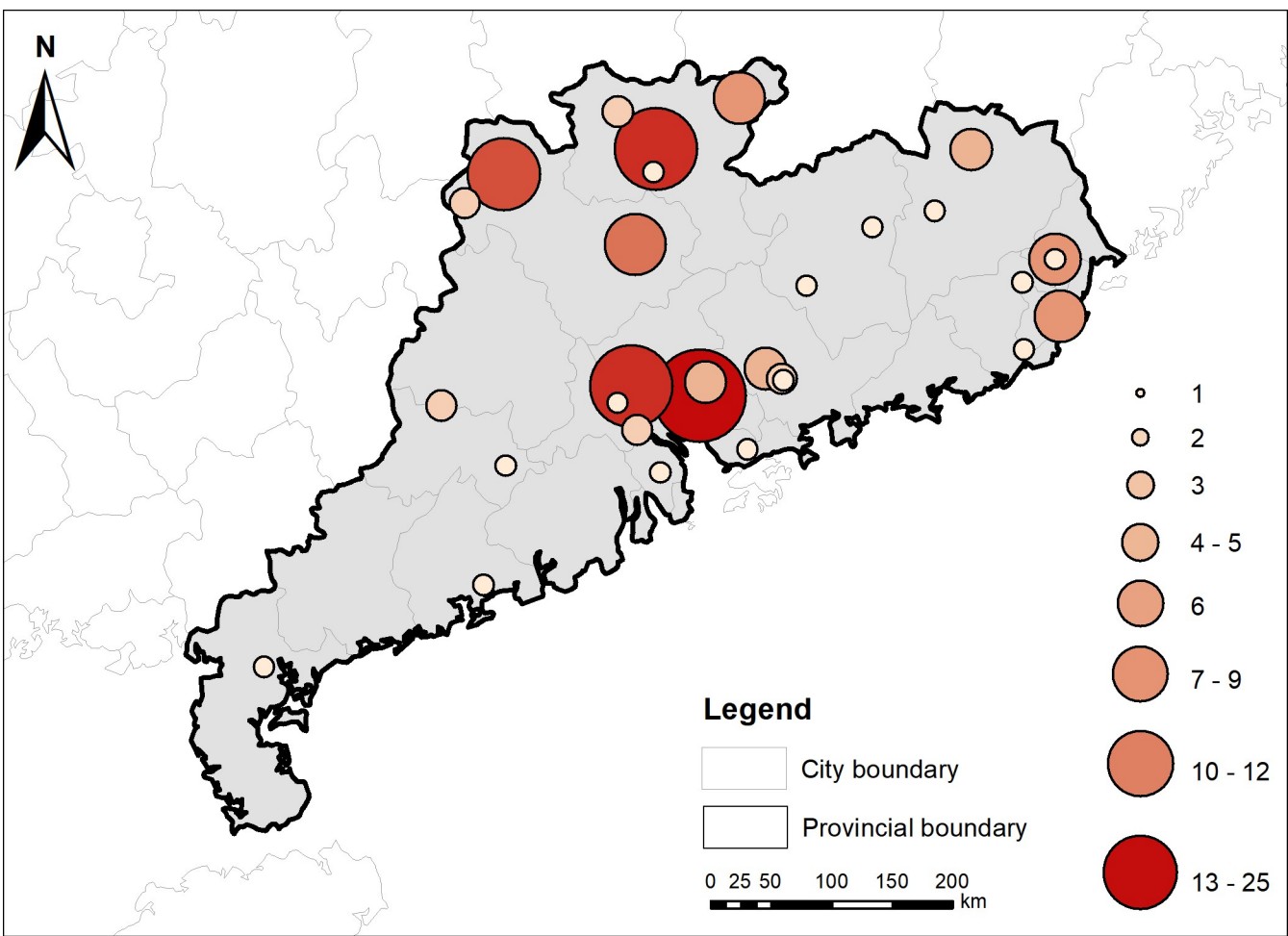

**Fig 5. Distribution map of Guangdong poets in the *Complete Song Poetry*.** Coordinate information of the poet's native place see S2 Dataset. The base map was obtained from Natural Earth [9]. Map credit: Xudong Hu.

Consequently, poets in Shaozhou are concentrated in Qujiang, as represented by Huang Zhongtong (986–1059), Yu Jing (1000–1064) and others. Chaozhou, located in the eastern part of Guangdong, was not a transportation hub. However, since Han Yu (768–824), a scholar-official in Tang Dynasty, was exiled to Chaozhou, he appointed a local scholar Zhao De as the head of the state school, bringing new thoughts to the style of Chaozhou scholars, making them eager to learn, and thus promoting the development of Chaozhou culture and education. Su Shi (1037–1101), a famous poet in in Northern Song Dynasty, recorded that:

"At first, the people of Chaozhou were not very knowledgeable, so Han Yu appointed a local scholar Zhao De to teach them. Since then, people in Chaozhou have attached great importance to cultural cultivation, and this trend has also spread to ordinary people, which has continued to this day and is known as good governance" [17].

Li Shichun (1585–1665) also wrote in his "Shuyuanji" (Records of the Academy):

"Before the Tang Dynasty, Chaozhou's reputation and educational level were not very high, and its cultural heritage was not rich enough. However, the beautiful natural scenery and

magical atmosphere here are hidden in dense forests, wetlands, and cities. Since Changli, courtesy name of Han Yu, became the governor of Chaozhou and Zhao De became his mentor, scholars here began to realize the importance of learning. As a result, the landscape of Chaozhou has been revitalized" [18].

Under the influence of previous sages, Chaozhou of Song Dynasty also saw the emergence of a group of poets, such as Lu Tong (1023–1094) and Wang Dabao (1094–1170), both of them being known as the member of "the Eight Sages of Chaozhou". Similar to Chaozhou, Lianzhou also underwent a significant shift in literary style after Liu Yuxi (772–842), an essayist and poet in Tang Dynasty, was exiled here. As a result, a new group of poets emerged in Lianzhou during the Song Dynasty, represented by Hu Junfang, Wu Shifan, Liao Meng, Ouyang Jing and others.

**The distribution of poets in Ningbo, Luoyang and Meishan, and their formation causes.**   Ningbo, also known as Mingzhou in the Northern Song Dynasty and Qingyuanfu in the Southern Song Dynasty, governed Yinxian, Cixi, Fenghua, Dinghai, Xiangshan, Changguo and other counties. Its unique geographical location was recorded in *Qiandao Siming Tujing* (Siming Map and Classics in Qiandao reign) as following:

"Mingzhou is in the eastern part of Yue (a region in ancient China, mainly located in the Zhejiang.). If you look at the map, it is in a rather remote corner. Although not a metropolis, it is an important hub for maritime transportation. It is bounded by Fujian and Guangzhou in the south, faces Japan in the east, and is neighbored by Koera in the north. Numerous merchant ships pass by, making it a flourishing commercial hub. Heading east to Dinghai, you will encounter Jiaomen and Hudun mountain, making Mingzhou an important rendezvous point in the southeast as well" [19].

This means that Ningbo is located in a coastal area with ease access to Fujian and Guangdong provinces internally, and to Japan and North Korea externally, making its sea transportation particularly convenient. This unique location makes Ningbo a convenient sea transportation port. Similar to Guangzhou, Ningbo's convenient transportation and developed ports provide favorable conditions for overseas trades. In 999 AD, Mingzhou Shibosi (Mingzhou Maritime Trade Supervisorate) was established, which further standardized port shipping and promoted trade activities. This resulted in a substantial increase in shipping revenue for Ningbo, laying a solid economic foundation for the flourishing of its culture and the rise of poet number. Notable and representative poets from Ningbo during the Song Dynasty include the "Five Gentlemen of Qingli" in the early Northern Song Dynasty, namely Yang Shi, Du Chun, Lou Yu (1008–1078), Wang Zhi (986–1055), and Wang Shuo (1010–1085). Additionally, the "Four Gentlemen of Yongshang" in the Southern Song Dynasty, including Yang Jian (1141–1225), Yuan Xie (1144–1224), Shu Lin (1136–1199), and Shen Huan (1139–1191), along with Lou Yao (1137–1213) and Wu Wenyin (1200–1260), were prominent representatives of poets in Ningbo.

During the Tang Dynasty, Luoyang, serving as a provisional capital, formed the "east capital" and "west capital" alongside capital Chang'an. However, since the Five Dynasties period (907–960), Chang'an lost its status as the capital, and the political and cultural centers in the north shifted to Kaifeng. Nonetheless, Luoyang remained highly esteemed by the people of the Northern Song Dynasty due to its rugged terrain, which acted as a defense against northern invasions. Fan Zhongyan (989–1052), a Chancellor and military strategist in Song Dynasty, remarked: "During times of peace, one can reside in the bustling and well-connected Kaifeng as it benefits from its convenient transportation. However, during war times, Luoyang's

formidable mountains and rivers offer a stronghold to protect the Central China" [20]. Thus, Luoyang, known as west capital during the Northern Song Dynasty, continued to be a provisional capital. Due to its proximity to Kaifeng, many officials from Kaifeng bought land and built houses in Luoyang as a place for leisure and recuperation. Some even constructed tombs there, envisioning it as their final resting place. This provided a solid foundation for the flourishing culture of Luoyang during the Song Dynasty. Li Gefei(?-1106), an academic professor from the Song Dynasty and father of the female poet Li Qingzhao(1084–1155), documented nineteen gardens in his "Famous Gardens in Luoyang" [21]. These private gardens played a crucial role in shaping Luoyang's distinctive garden culture and creating an ideal cultural environment for the emergence of local poets. Notable literary figures in Luoyang during this time included Chen Yuyi (1090–1138), Zhu Dunru (1081–1159), Fu Bi (1004–1083), Fu Zhirou (1084–1156), Shao Bowen (1057–1134), Cheng Yi (1033–1107), Cheng Hao (1032–1085), and other influential politicians, thinkers, and poets.

Meishan in Sichuan was named Meizhou during the Song Dynasty, located within the Sichuan Basin. This Basin was densely surrounded by numerous mountains, forming a natural barrier to resist enemies from the perspective of geographical location and topography, and people's lives here can be self-sufficient. Consequently, the Sichuan Basin has been an ideal place to avoid war since ancient times. During the war in the late Tang Dynasty, a large number of northerners and their families migrated to Sichuan to avoid the war. According to statistics, Meizhou accommodated the largest number of immigrants among all Sichuan province, which laid a strong population foundation for prosperity of Meishan culture. The developed academies and imperial examinations eventually formed the unique Meishan immigration phenomenon in the Song Dynasty [22], creating a group of well-known Meishan poets represented by Su Shi.

## Conclusion

In summary, this article analyzes the distribution of over 9000 poets in the *Complete Song Poetry* and finds that the poets of the Song Dynasty mainly formed a "three core" distribution pattern around Zhejiang, Henan and Sichuan. The distribution of poets in Zhejiang and Fujian provinces always held a leading position during the Song Dynasty. Guangdong, despite being far from the capital, also shows a polarized distribution due to its maritime trade. Ningbo, Luoyang and Meishan also have a larger distribution of poets due to their unique geographical locations. Through visual analysis of the geographical distribution of poets in *Complete Song Poetry*, this article presents a clearer and more accurate geographical distribution of poets in the Song Dynasty, revealing the relationship between the prosperity of Song Dynasty culture and economic development.

## Supporting information

**S1 Dataset. Database of poets' native places in the *Complete Song Poetry*.**
(XLS)

**S2 Dataset. Database of Gangdong poets' native places in the *Complete Song Poetry*.**
(XLS)

**S1 File. Supporting information for references, citations and figures.**
(PDF)

## Acknowledgments

We sincerely acknowledge for Prof. Zhibing Zhang's advice for our manuscript. We also thank Xuewei Zhao and Xueli Feng for their assistance during the preparation of our manuscript.

## Author Contributions

**Conceptualization:** Enhai Lei.

**Data curation:** Xudong Hu.

**Formal analysis:** Xudong Hu.

**Funding acquisition:** Enhai Lei.

**Investigation:** Xudong Hu.

**Methodology:** Enhai Lei.

**Project administration:** Enhai Lei.

**Resources:** Xudong Hu.

**Software:** Xudong Hu.

**Supervision:** Enhai Lei.

**Validation:** Enhai Lei.

**Visualization:** Xudong Hu.

**Writing – original draft:** Xudong Hu.

**Writing – review & editing:** Enhai Lei, Xudong Hu.

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
