## [Decision Letter · Decision Letter 0]

29 Jul 2024

PONE-D-24-23728Visual Analysis of Geographical Distribution of Poets in Song China Based on Complete Song PoetryPLOS ONE

Dear Dr. Lei,

Thank you for submitting your manuscript to PLOS ONE. After careful consideration, we feel that it has merit but does not fully meet PLOS ONE’s publication criteria as it currently stands. Therefore, we invite you to submit a revised version of the manuscript that addresses the points raised during the review process.

We look forward to receiving your revised manuscript.

Kind regards,

Dawit Dibekulu, PhD

Academic Editor

PLOS ONE

 [This research was funded by the Gansu Education Department (2022CYZC-09).].  

Reviewers' comments:

Reviewer's Responses to Questions

**Comments to the Author**

1. Is the manuscript technically sound, and do the data support the conclusions?

Reviewer #1: Partly

Reviewer #2: Partly

2. Has the statistical analysis been performed appropriately and rigorously? 

Reviewer #1: No

Reviewer #2: Yes

3. Have the authors made all data underlying the findings in their manuscript fully available?

Reviewer #1: Yes

Reviewer #2: Yes

4. Is the manuscript presented in an intelligible fashion and written in standard English?

Reviewer #1: Yes

Reviewer #2: No

5. Review Comments to the Author

Reviewer #1: The manuscript titled "Visual Analysis of Geographical Distribution of Poets in Song China Based on Complete Song Poetry" offers a comprehensive analysis of the spatial distribution of poets during the Song Dynasty. By leveraging the "Complete Song Poetry" collection and GIS visualization techniques, the authors have provided valuable insights and empirical data for the field. Although the study is well-executed and presented, there are opportunities to improve clarity, rigor, and overall quality.

Detailed Feedback:

Technical Soundness and Data Support for Conclusions

Technical Soundness: Partially

The study utilizes GIS software and a comprehensive dataset, which is a commendable approach. However, the methodology section lacks sufficient detail on data processing, GIS visualization steps, and statistical analysis. Inclusion of this information would enhance the technical rigor and reliability of the study.

Suggestions: Add detailed descriptions of the GIS data processing methods, statistical tests used, and how data accuracy was ensured.

Introduction Section Review

Current Introduction:

Paragraph 1:

"China is a country of poetry, and after thriving of poetry in Tang Dynasty, the Song China (960-1279) saw another summit in the development of Chinese poetry. Thousands of poets were appeared during this time and scattered across China."

Comment: This opening sets the context well but could be smoother. It should also mention the importance of understanding the geographical distribution of these poets.

Paragraph 2:

"Studying the geographical distribution of poets in Song Dynasty can get insights into the cultural, economic, and political development of Song Dynasty society. From statistical and geographical distribution perspectives, the research works on the distribution of poets in the Song Dynasty have been performed previously in the academics, resulting many research achievements."

Comment: This paragraph introduces the significance of the study and acknowledges prior research. However, it could transition more smoothly by linking the historical context to the need for this study.

Paragraph 3:

"As far back as the 1990s, Zeng Daxing's book 'Geographical Distribution of Chinese Literary Masters of Previous Dynasties' [1] presented a statistical analysis of the geographical distribution of ancient Chinese literary masters. In chapter 6 of this book, the geographical distribution patterns of literary scholars in Song, Liao, and Jin Dynasties along with their formation reasons were discussed based on the number of writers’ native places recorded in Tan Zhengbi’s 'Dictionary of Chinese Literati' [2], and 1168 poets were involved in total."

Comment: This paragraph delves into specific prior works. It transitions from general significance to specific examples, which is logical but could benefit from a clearer linkage.

Paragraph 4:

"In 2003, Wang Zhaopeng and Liu Xue published a paper entitled “Statistical Analysis of Poets in Song Ci” [3], in which a statistical analysis of 880 poets in Song Dynasty was performed, analyzing their geographical and temporal distributions. They concluded that the development of Song Ci, a special type of poem in Song Dynasty, was intimately linked to the prosperity of Song poetry."

Comment: This continuation of prior works is consistent with the previous paragraph. Ensure smooth transition by summarizing how these works build a foundation for the current study.

Paragraph 5:

"In Lou Xinxing’s article “Geographical Distribution of Authors in Complete Song Iambic Verse Collection and the Cause” [4], a statistical and visual analysis about the distribution of poets in Complete Song Ci at the provincial and county levels was performed, with a particular focus on poets from Zhejiang province."

Comment: Maintains consistency in reviewing past research. Ensure to link it back to the current study's objectives for better coherence.

Paragraph 6:

"Additionally, Wang Xiang's 'Analysis of the Geographical Distribution of Poets in Northern Song Dynasty and Their Significance in Literary History' [5] recorded the geographical distribution of 975 poets in the Northern Song Dynasty based on Complete Song Poetry, which was compiled by the Center for Ancient Chinese Classics and Archives of Peking University and published by Peking University Press in 1998. He divided the Northern Song Dynasty into four stages from a literary history perspective to discuss the geographical distribution of poets, providing insights into the literary prosperity and cultural center of the Song Dynasty."

Comment: Another specific example of prior work. Transition is logical but could be enhanced by summarizing the cumulative insights these studies provide for the current research.

Paragraph 7:

"In a word, the previous studies on the geographical distribution of poets during the Song Dynasty involved about a thousand individuals. In fact, the Complete Song Poetry documents over 9000 poets from the Song Dynasty in its total of 72 volumes, offering a comprehensive overview of Song Dynasty poets. Therefore, studying on the geographical distribution of all poets in the Complete Song Poetry is essential for a comprehensive understanding of poets' distribution in Song Dynasty, and then offering more accurate insight into the regional characteristics of literature in Song Dynasty."

Comment: This paragraph effectively summarizes the gaps in the existing literature and sets the stage for the current study. The transition here is smooth and logical.

Paragraph 8:

"Given the massive number of poets in the Complete Song Poetry, the Complete Song Poetry Analysis System, [6] which is developed by the Department of Chinese Language and Literature of Peking University in 2005, will be instrumental in collecting and processing the extensive and intricate data of poets with clear native place records, enabling quantitative analysis of their geographical distribution through statistical method. In particular, GIS software will be used herein to visually display the geographical distribution and density of the poets in Complete Song Poetry to explore the characteristics of the geographical distribution of poets and the formation reasons behind it, and thus making progress for the research works on the investigation of the geographical distribution of poets from the Song Dynasty."

Comment: This final paragraph clearly outlines the methodology and tools used in the current study, effectively transitioning from the literature review to the research methodology.

Suggested Improvements for Transitions:

Smooth Transitions: To enhance the smoothness of transitions, consider adding brief summary sentences at the end of each paragraph that link to the next one. This can help guide the reader through the narrative logically.

Consistent Linking: Ensure each paragraph explicitly connects back to the central research question of the current study. This can be done by emphasizing how each piece of prior research contributes to understanding the broader topic.

Cohesive Flow: Use transition phrases like "Building on these findings," "In continuation," "Expanding upon," and "Furthermore," to improve the flow between paragraphs.

Statistical Analysis

Statistical Analysis: No

The manuscript should include comprehensive descriptions of the statistical methods utilized to validate the findings. Thorough statistical analysis is imperative for establishing the credibility and robustness of the results.

Suggestions: In your analysis, please provide a detailed description of the statistical methods utilized, including the specific tests applied, significance criteria, and data handling procedures. Additionally, discuss the validation of results and any replication or controls utilized in the study.

Data Availability

Data Availability: Yes

The authors have ensured that all pertinent data supporting their findings are included in the manuscript and its accompanying information files. This practice aligns with PLOS ONE's data policy, which emphasizes transparency and the ability to reproduce the results.

No further action needed.

Presentation and Language

Presentation and Language: Yes

The manuscript demonstrates strong structural organization and employs standard English conventions. Nonetheless, it contains minor grammatical and typographical errors that warrant attention to enhance overall readability.

Suggestions: Conduct a thorough proofread to correct grammatical errors and awkward phrasings. Examples include:

Abstract: "With the time going on" → "Over time"

Introduction: "after thriving of poetry in Tang Dynasty" → "after the thriving of poetry in the Tang Dynasty"

Results: "data from Fig 1 reveals" → "data in Fig 1 reveals"

Implications and Discussion

Implications and Discussion: Needs Improvement

The results are well organized, but the discussion section needs to delve deeper into the wider implications of the findings. This should encompass their influence on our comprehension of Song Dynasty society and the variables affecting the geographic spread of poets.

Suggestions: Discuss the socio-political and economic contexts that may have influenced the distribution patterns. Address any limitations of the study and potential biases in the data.

Clarity of Figures

Clarity of Figures: Needs Improvement

The visual representations, such as figures and maps, aid in articulating the discoveries, although some require enhanced clarity and readability. It is essential to augment the resolution and labeling in order to proficiently communicate the data.

Suggestions: Please enhance the clarity and readability of all figures by increasing the resolution, improving labeling, and providing more detailed captions to guide the reader.

Additional Comments:

Figures and Visualizations:

Please ensure that the figures and maps are accompanied by comprehensive and descriptive captions, along with in-text explanations to maximize their utility. Additionally, make sure that each figure is unambiguously referenced and discussed within the main body of the text.

References:

The references are cited correctly and written in a standard format. However, make sure all sources are accurately cited and consistently formatted throughout the manuscript.

However, ensure all references consistently include:

Author(s)

Year of publication

Title of the work (italicized for books and journal names, with proper capitalization)

Publisher or journal name

Volume and issue number (for journal articles)

Page numbers (if applicable)

Make sure to check for any inconsistencies in the format throughout the manuscript to maintain uniformity.

Ethical Considerations:

The manuscript adheres to ethical standards, with clear statements on data availability and no competing interests declared.

Conclusion:

The inclusion of figures and maps is beneficial; however, augmenting their utility with more detailed captions and in-text explanations would be advantageous. It is important to ensure that each figure is explicitly referenced and thoroughly discussed in the text.

Recommendation:

Accept with minor revisions.

To improve their manuscript, the authors should consider addressing the specific points mentioned in order to bolster the clarity, robustness, and overall quality. This will result in a more impactful contribution to our understanding of the cultural and historical landscape of the Song Dynasty.

Reviewer #2: The manuscript is with major flaws, geographical representations of poets does not describe the significance of this research. There should be rationale behind conducting this type of research. Problem statement and research questions should be included. In-text citations and references are in pathetic condition. Methodology is not clear.

6. PLOS authors have the option to publish the peer review history of their article (what does this mean?). If published, this will include your full peer review and any attached files.

Reviewer #1: **Yes: **Mohanad Husni Al Jbour

Reviewer #2: No

---

## [Author Response · Author response to Decision Letter 0]

22 Aug 2024

Dear Reviewers,

Thank you very much for giving us a valuable opportunity to revise our manuscript, and we really appreciate your significant review assistance for handling our manuscript and sharing the reviewer's comments on our manuscript (ID: PONE-D-24-23728, entitled: “Visual Analysis of Geographical Distribution of Poets in Song China Based on Complete Song Poetry”).

We have carefully studied reviewers’ comments and tried our best to make point-to-point revisions which are marked in red in our marked-up copy. We believe that the revised manuscript would be more reasonable.

The revision reports are listed as following in detail. We hope that the changes can satisfy your requirements. If you have any question, please contact us at any time..

Sincerely yours,

Enhai Lei

School of Chinese Languages and Literatures, Lanzhou University, Lanzhou, P. R. China

---

## [Editor Report · Decision Letter 1]

26 Aug 2024

Visual analysis of geographical distribution of poets in Song China based on Complete Song Poetry

PONE-D-24-23728R1

Dear Dr. ,

We’re pleased to inform you that your manuscript has been judged scientifically suitable for publication and will be formally accepted for publication once it meets all outstanding technical requirements.

Kind regards,

Dawit Dibekulu, PhD

Academic Editor

PLOS ONE
---

## [Editor Report · Acceptance letter]

28 Aug 2024

PONE-D-24-23728R1 

PLOS ONE

Dear Dr. Lei, 

I'm pleased to inform you that your manuscript has been deemed suitable for publication in PLOS ONE. Congratulations! Your manuscript is now being handed over to our production team.

Kind regards, 

on behalf of

Dr. Dawit Dibekulu 

Academic Editor

PLOS ONE